# Glycerol-based deep eutectic solvents for efficient and reversible iodine uptake from vapour phase

Daniele Motta [1,5], Saïd Mondahchouo[2,5], Stefano Nejrotti[1,3], Carlotta Pontremoli[1], Claudia Barolo [1,3], Alessandro Damin [1,6] ✉ & Matteo Bonomo [1,4,6] ✉

The efficient capture and recovery of radioactive iodine ($I_2$) is crucial for nuclear safety and environmental protection. In this study, we propose some glycerol (Gly)-based deep eutectic solvents (DESs) as novel and promising solvents for iodine uptake from the vapour phase. Depending on the cholinium salt ($Ch^+$) selected as hydrogen bond acceptor (iodide or chloride) and its relative ratio with respect to glycerol (1:2 or 1:3), the $I_2$ uptake approaches or even overcomes the 300 ms% with ChCl:Gly 1:2 showing the best performance with $4\,g\,g^{-1}$ after 24 h. Moreover, the captured $I_2$ can be effectively (almost 80%) released showcasing the potential of DESs as iodine sponges. Finally, the exploitation of Raman spectroscopy proves the speciation of I-based species within the DES, paving the way for further engineering of these systems. Our results highlight Gly-based DES as a sustainable and cheap solvent for effective and semi-reversible $I_2$ sponges.

Despite the remarkable progress in energy production from renewable sources (*e.g.* photovoltaic, eolic…), nuclear energy still accounts for up to 9% of the world's total electricity demand[1]. However, the electrical energy coming from nuclear power plants relies on the fission of uranium-235, whose major by-products are iodine isotopes, *i.e.* [129]I and [131]I. The latter (usually in the form of $I_2$ vapours) could not get simply released in the atmosphere being highly environmentally harmful and showing not negligible residual radioactivity[2]. As such, to improve the greenness of nuclear energy and to classify it as a sustainable energy source, it is an urgent need to find reliable, reversible, and cost-effective strategies to sequestrate $I_2$ vapours. Nowadays, state-of-the-art absorbents are solid and highly porous materials which can chemically or physically sequestrate $I_2$[3]. Indeed, porous inorganic absorbents[4] and, more recently, organic materials[5], have shown remarkable results in terms of both capture efficiency and material stability, with the uptake ability depending strongly on the availability of adsorption sites. As a further step, some liquids (*e.g.* ionic liquids, ILs) have been coupled with solid adsorbers leading to increased uptake values. For example, Zhang et al.[6] proposed polymer/pyridine gels which led up to $3.0\,g\,g^{-1}$ of iodine uptake thanks to the nitrogen binding sites. Following this N-rich strategy, the storage capability of the system can be improved up to

$6.94\,g\,g^{-1}$ by incorporating the IL in a melamine/formaldehyde foam[7]; similarly, Liao et al.[8] demonstrated that the presence of amine groups coupled with a microporous-rich morphology of hexaphenylbenzene-based conjugated microporous polymers (HCMP) ensure a high-affinity toward iodine (336 ms% of uptake). Some of us reported the use of a low surface area quinoid-thiophene-based covalent organic polymers (COPs) ensuring a 428 ms% uptake and pinpointing how a thoughtful chemical design of the absorber could counter-balance its sub-optimal morphological properties[9]. As far as we are aware, the highest uptake values have been reported by Xie et al.[10] who proposed 1,3,5-tris(4-aminophenyl) benzene (TAPB)- and tris(4-formyl phenyl)amine (TFPA)-based covalent organic frameworks (COFs) approaching and even surpassing the limit of $8\,g\,g^{-1}$ (7.94 and $8.61\,g\,g^{-1}$, respectively); however, it requires a very long saturation time (4 days), posing some doubts on their practical application taking into account the $T_{50}$ of [131]I (*i.e.* 8 days). Slightly lower uptake values (700 ms%) but with a more reasonable saturation time (24 h) have been reached by Sen et al.[11], designing ionic (*i.e.* charged) porous organic polymers (POPs).

Notwithstanding these achievements, the uptake capacity of porous solid adsorbents seems to have reached a plateau. Moreover, porous-based

[1]Department of Chemistry, NIS Interdepartmental Centre and INSTM Reference Centre, University of Turin, Turin, Italy. [2]Laboratory of Analytical Electrochemistry and Materials Engineering, University of Yaoundé I, Yaoundé, Cameroon. [3]Institute of Science, Technology and Sustainability for Ceramics (ISSMC-CNR), Faenza, Italy. [4]Department of Basic and Applied Science for Engineering, La Sapienza University of Rome, Rome, Italy. [5]These authors contributed equally: Daniele Motta, Saïd Mondahchouo. [6]These authors jointly supervised this work: Alessandro Damin, Matteo Bonomo. ✉e-mail: alessandro.damin@unito.it; matteo.bonomo@uniroma1.it

materials have plenty of concerns, particularly in terms of sustainability, mainly due to the quite complex and energy-demanding synthetic procedure. On the other hand, although liquid adsorbents (mainly ILs) have been proposed as additives to modulate the uptake capacity of solid counterparts, their exploitation as stand-alone iodine sponges is still limited. In this context, remarkable iodine storages of 14.1, 15.8, and 17.5 g g$^{-1}$ are achievable with different imidazolate ILs[12]; however, these systems do not behave like sponges, since triggering the release of iodine is not possible even under harsh conditions (10 h at 100 °C), due to the strong chemical bond established between cations and polyiodide species. Beyond this concern, ILs are also relatively expensive and difficult to purify after synthesis. Moreover, the toxicity of the commonly used imidazolium/imidazolate cations for IL synthesis is currently a big concern[13]. As a promising and greener alternative, deep eutectic solvents (DESs) have recently been proposed due to their simpler synthesis procedure and their environmental benignity[14]. Here, the use of halogen-containing DESs should be advantageous to (reversibly) sequestrate I$_2$ due to the predicted strong halogen bonding (XB) established between iodine and the X$^-$ anion, both in liquid and/or vapour phase. Screening the literature, the few existing examples are all related to the DES-assisted liquid-liquid extraction of I$_2$ for contaminated solvents, as first proposed by Yan et al. with several ILs[15]. As a paradigmatic example, Li et al.[16] screened a series of DES and DES-like systems to extract iodine from cyclohexane, finding choline iodide/methyl urea (ChI/MU) as the most effective one, being able to quantitatively extract I$_2$ (uptake 0.99 g g$^{-1}$) in 5 h and proposing the formation of XB as the main driving force. Albeit corroborated by DFT calculations, some doubts arise on the preservation of the DES structure when mixed with I$_2$-loaded cyclohexane, since the interaction of the latter with the mixtures has not been considered. Indeed, it is worth mentioning that in a more recent work[17], DES analogues have demonstrated a certain affinity towards cyclohexane, proving effective as volatile organic compound (VOC) absorbents.

Notwithstanding other reports that followed this approach[18–20], we could not find any paper describing the DES-mediated iodine uptake from vapour phase. Throughout this communication, the exploitation of choline iodide-glycerol (ChI-Gly) and choline chloride-glycerol (ChCl-Gly) DESs as very promising gaseous iodine sponges is proposed for the first time, showing an uptake capacity of up to 4 g g$^{-1}$. This value suggests that the uptake mechanism is likely not (limited to) the I$_2$-X$^-$ association, as also corroborated by the outperforming of iodine-based DES by chloride-based ones. The exploitation of Raman spectroscopy allows us to shed light on this, suggesting a dramatic active role of the hydrogen bond donor (glycerol) in the uptake mechanism.

## Results and discussion
The iodine absorption efficiency of a DES is expected to be strongly correlated to the chemical nature of hydrogen bond acceptor (HBA) and hydrogen bond donor (HBD), and especially to the halogen atom of the HBA. Here, we formulated three types of glycerol-based DES (Fig. 1a) differing in the HBA nature (choline iodine or chloride) and the HBA/HBD molar ratio (1:2 or 1:3). The thoughtful selection of DES ensures a focus on the role of the halogen: indeed, the likely formation of halogen bonds could ensure the uptake of a great amount of iodine vapour.

### Gravimetric iodine uptake
DES1 (ChCl-Gly 1:2), DES2 (ChCl-Gly 1:3), DES3 (ChI-Gly 1:3), and glycerol were used to capture vapour iodine using the setup described in the Methods. From Fig. 1b (for the complete dataset, see Supplementary

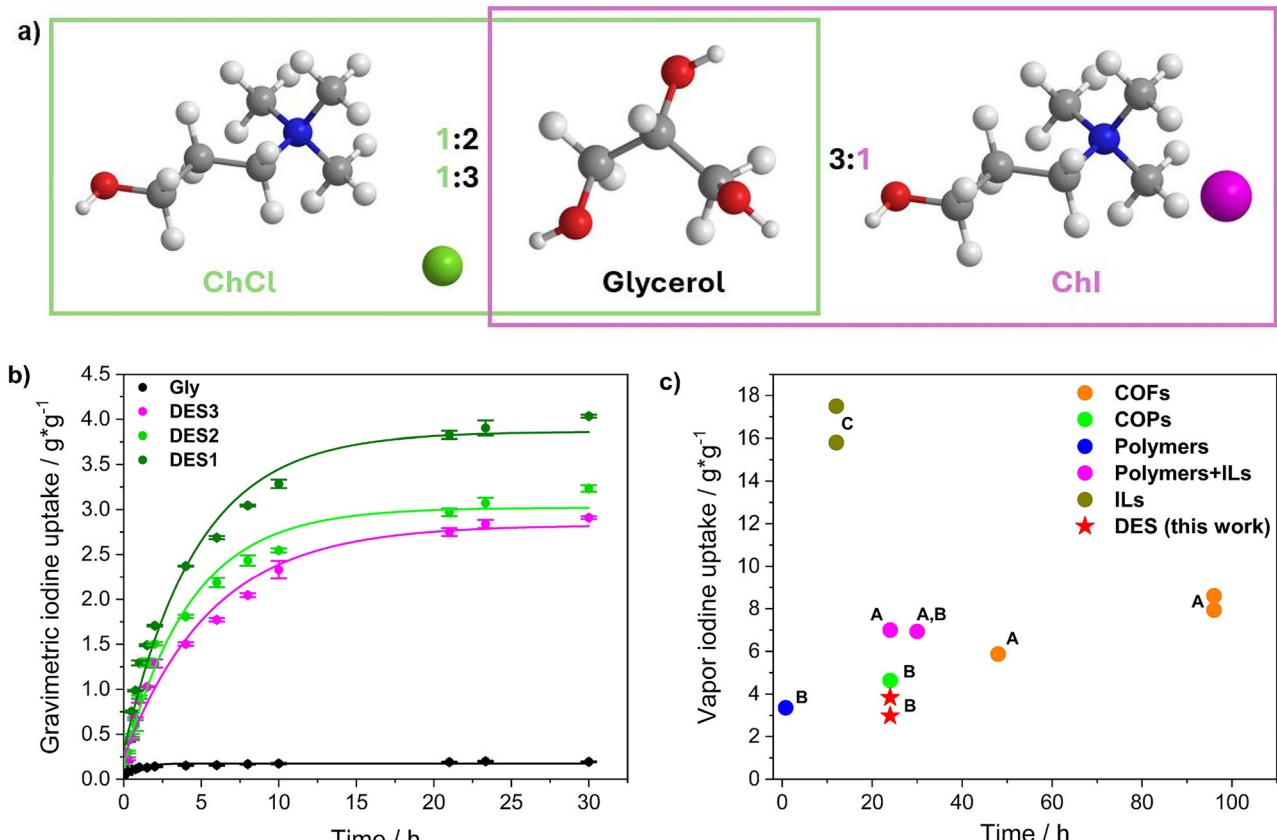

**Fig. 1 | Gravimetric iodine uptake and comparison with the state of the art.**
**a** Schematic representation of the proposed DESs for vapour iodine uptake.
**b** Gravimetric iodine uptake on DES1, DES2, DES3, and glycerol from 0 to 30 h with error bars (standard deviation). **c** Comparison of vapour iodine uptake efficiency and time between the proposed DESs and the best-performing materials reported in the literature (Supplementary Table 1); A = regenerated in solution, B = reversible after heating, C = non reversible.

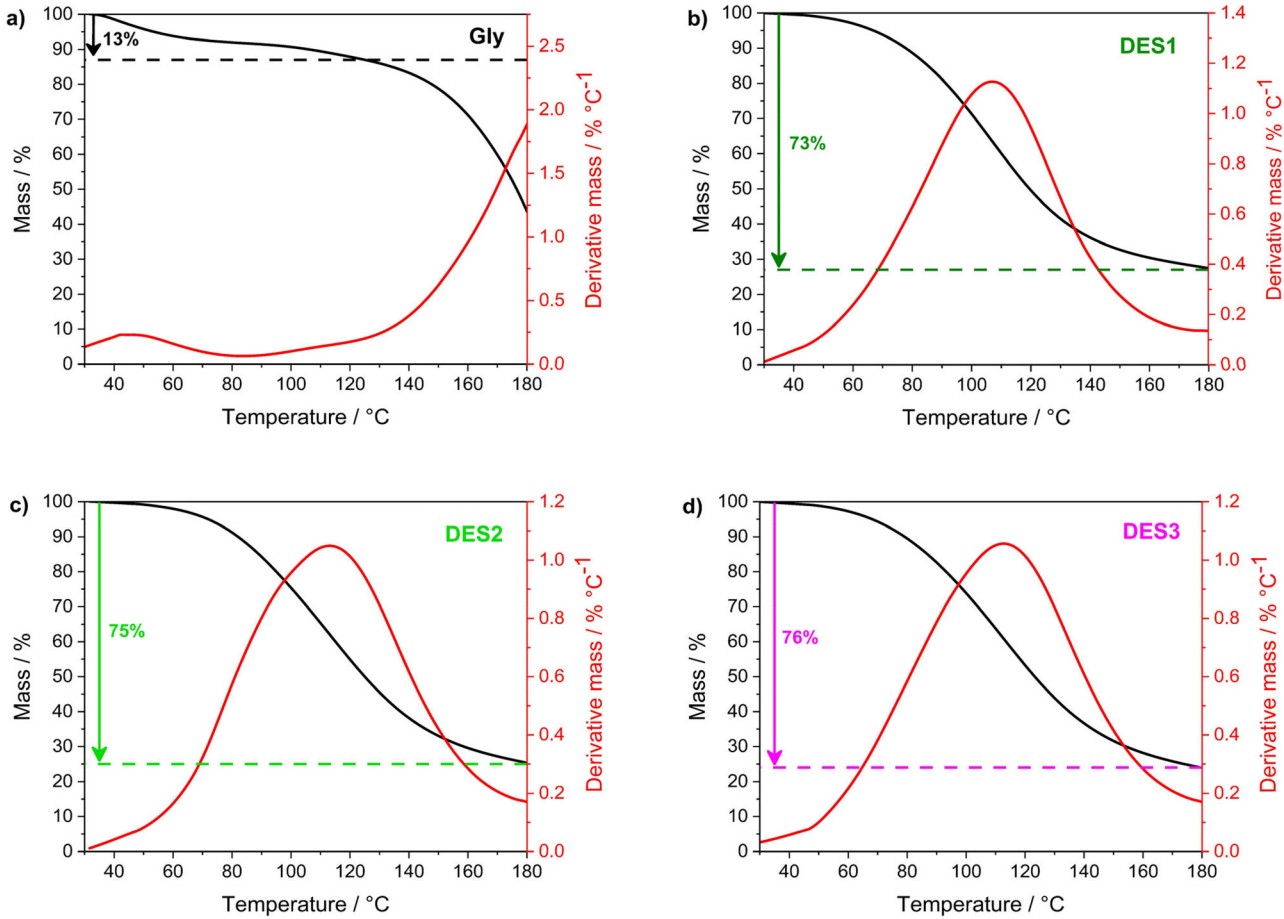

**Fig. 2 | TGA and DTG analyses of iodine-loaded.** TGA (black curves) and DTG (red curves) analyses of iodine-loaded **a** glycerol, **b** DES1, **c** DES2, **d** DES3. The arrows indicate the mass percentage of iodine loss in each system.

Data 1–2), it can be noticed that DES1, DES2, and DES3 provide a fast uptake, with a massive absorption, reaching a plateau within 21 h. On the other hand, the uptake capacity of pure glycerol is extremely low compared to the eutectic mixtures, pinpointing the active role of the HBA in driving the $I_2$ capture. Once reached the plateau of the $I_2$ uptake, all the samples showed a quasi-solid gel-like behaviour, which is even more evident when the samples are cooled down to RT. This is fairly ascribed to the establishment of new Iodine-based interaction between DES moieties (see Raman section). The high-energy nature of the latter leads to extremely stable $I_2$-loaded sponges which lose only an amount between 3% and 6% of mass after more than 1 month (*i.e.* 42 days) of storage under ambient conditions. As shown in Supplementary Fig. 1, almost all the weight is lost within the first 4 h of storage, likely due to the evaporation of the loosely interacting $I_2$ at the DES exposed surface. If on the one hand, the establishment of highly energetic interactions is surely an added value in terms of long-term stability toward safe and reliable industrial application, it would also require some energy to allow an effective de-loading of the iodine (*vide infra*).

The amount of iodine sequestered by pure glycerol could be considered as a baseline for the maximum physical entrapment of the vapour within the solvent, even if a specific interaction between the $I_2$ and the hydroxyl group could not be excluded a priori. The highest amount of iodine captured by DES1, DES2, and DES3 compared to glycerol could account for the highest interaction energy between these adsorbents and $I_2$, supported by the formation of XB during the capture process. This strong interaction is further reinforced by the weaker electrostatic interaction between choline and $Cl^-$ and $I^-$ thus, resulting

in a scaffold with high affinity for $I_2$. By virtue of experimental data fitting through exponential asymptotic function (Supplementary Table 2), the extrapolated amounts of iodine captured by DES1, DES2, and DES3 are 3.86 g g$^{-1}$, 3.02 g g$^{-1}$ and 2.83 g g$^{-1}$, respectively. These values are significantly higher than the 0.17 g g$^{-1}$ for pure glycerol and are comparable to, or even surpassing, the performance of the most effective absorbents used for iodine vapour uptake in literature (Fig. 1c and Supplementary Table 1), particularly when considering the reversibility as well. Although water absorption may influence gravimetric measurements, its impact is considered negligible in the present case. Based on previous findings[21], the theoretical maximum water absorption capacity for these kind of systems is approximately 10% by mass, which would represent only 2–3% of the total uptake observed (*i.e.* a value within the experimental uncertainty). Furthermore, the saturation of sample containers with iodine vapours and the kinetic behaviour observed suggest limited room for significant water absorption.

## Thermal analysis

Albeit the relatively high iodine uptake ensured by glycerol-based DESs, to have a practical industrial application as iodine sponges, they should also show a controllable release of the $I_2$ under mild conditions. Therefore, we run some thermogravimetric analyses (TGA) to investigate the behaviour of $I_2$-loaded DESs under thermal conditioning (Fig. 2). Pure glycerol, used as a reference (Fig. 2a), shows two small mass losses at 50 °C and 110 °C both ascribable to the loss of physically trapped or weakly interacting iodine. The overall mass loss is around 13% in very good agreement with the uptake data. The sudden drop at T > 140 °C is ascribable to the thermal degradation

of the glycerol, which occurs between 200 °C and 213 °C, both in its pure form and within the mixtures (Supplementary Fig. 2).

Quite surprisingly, the thermogravimetric curves of all the $I_2$-loaded DESs show an extremely similar profile (Fig. 2b–d), with a quite broad mass loss centred at around 120 °C and overall mass losses of around 75%. The latter could be imputed to molecular iodine release, accounting for an iodine uptake around $3 \, g \, g^{-1}$. This value is in extremely good agreement with the one measured during the uptake test for both DES2 and DES3, but quite lower for DES1 (see Fig. 1b). A closer inspection of the derivative of the curves (derivative thermogravimetry, DTG) reveals that the loss is made of (at least) two processes, as proved by the curve shapes which deviate from the ideal Gaussian curve.

Although TGA data prove a very promising reversibility in iodine uptake and release, they evidence a partial inconsistency between the amount of $I_2$ loaded and released, especially for DES1. Moreover, it does not

give us any definitive information on the uptake mechanism neither on the I speciation.

## Raman investigation

Through the combination of gravimetric and thermogravimetric analyses, we highlighted the capabilities of these DESs to store vapour iodine and (almost) reversibly release it after heating. It is evident that the structuring of the glycerol network with choline salts positively affects the uptake capability, as well as the establishment of multiple interactions that lead to sub-optimal release of the iodine stored. To elucidate the factors underlying these outcomes, the exploitation of Raman spectroscopy is pivotal to determine the iodine species involved in the uptake since molecular iodide and polyhalide anions are excellent Raman scatters[22]. If proving the presence of solid iodine ($I_2$) is relatively straightforward, due to the well-defined Raman-active mode at ≈180 cm$^{-1}$ [23,24], clarifying the formation of polyiodides in the mixtures is more challenging. Indeed, as deeply described over twenty years ago by Deplano et al.[25] and Svensson et al.[26], Raman spectra of these systems present three characteristic Raman features which could be consequently ascribed to three different species (Table 1): molecular iodine weakly bonded (intermolecular interaction) with electron donors (D-$I_2$), triiodide ($I_3^-$) or pentaiodide ($I_5^-$). Higher polyiodides (if present), e.g. heptaiodides, can be defined by using the aforementioned building blocks due to the minimal spectral differences from their lower counterparts.

**Table 1 | Typical Raman shifts for iodine species**

| Iodine species | Raman shifts |
|---|---|
| Solid $I_2$ | 180 cm$^{-1}$ |
| D–$I_2$ | 140–180 cm$^{-1}$ |
| $I_3^-$ | 143 cm$^{-1}$/110 cm$^{-1}$ |
| $I_5^-$ | 168 cm$^{-1}$/143 cm$^{-1}$/110 cm$^{-1}$ |

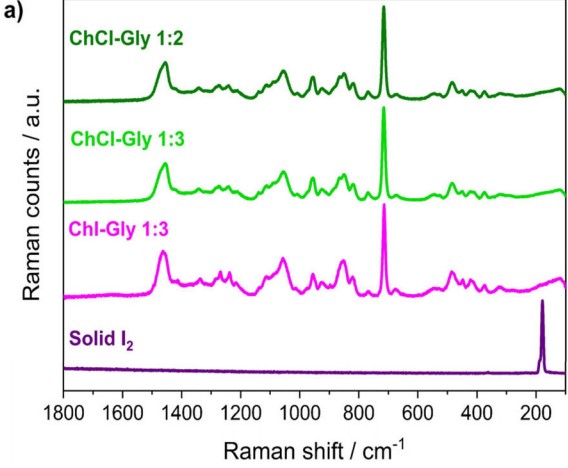

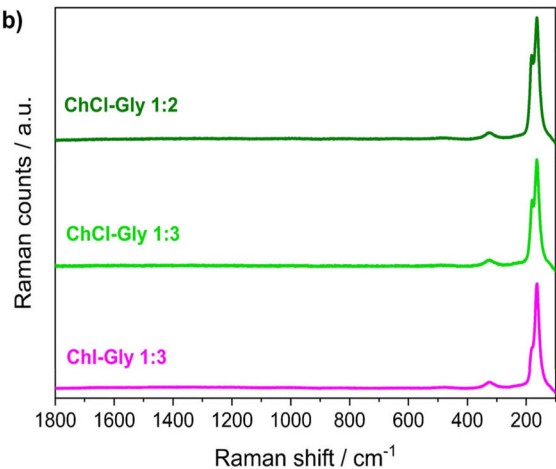

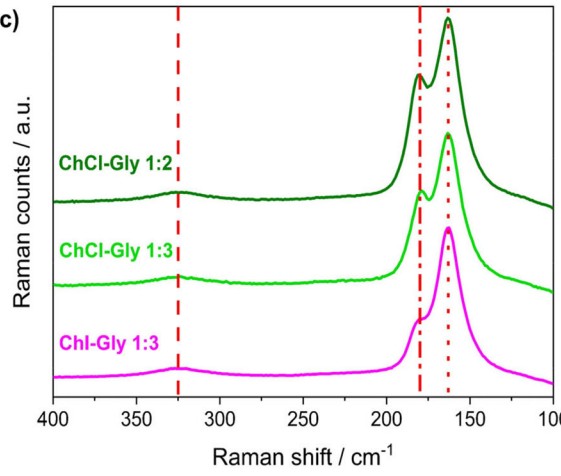

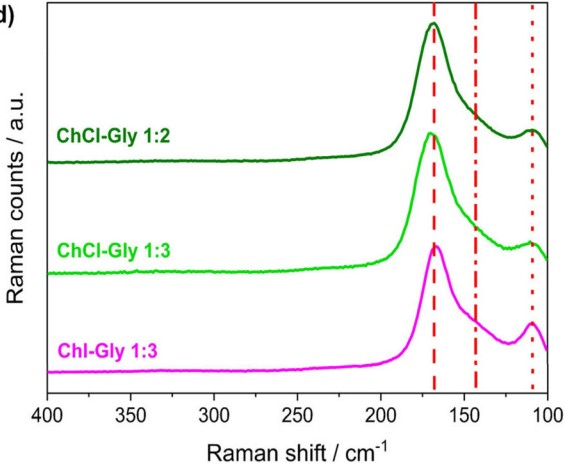

**Fig. 3 | Raman characterisation of iodine-loaded DESs.** Raman spectra of **a** DESs and solid $I_2$, **b** DESs after 24 h of iodine uptake in the region 100–1800 cm$^{-1}$, **c** DESs after 24 h of iodine uptake in the region 100–400 cm$^{-1}$. Red lines: dashed = 325 cm$^{-1}$, dash-dotted = 180 cm$^{-1}$, dotted = 163 cm$^{-1}$. **d** Loaded DESs after heating at 100 °C for 30 min. Red lines: dashed = 168 cm$^{-1}$, dash-dotted = 143 cm$^{-1}$, dotted = 109 cm$^{-1}$.

Essentially, more complex polyiodides can be understood as combinations of these simpler units, rather than entirely distinct molecular entities.

Raman spectra of each DES mixture (in their pure form in Fig. 3a) were collected after 24 h of iodine uptake ($4.00 \, g \, g^{-1}$, $2.79 \, g \, g^{-1}$, and $2.83 \, g \, g^{-1}$ for DES1, DES2, and DES3 respectively), which are consistent with the values obtained from the gravimetric curves). Due to the high intensity of the bands related to the iodine/polyiodide species, the characteristic Raman features of the solvent components (*i.e.* glycerol and choline) are no longer visible in the spectra after loading, as evidenced in Fig. 3b. Due to the flatness of the spectra at Raman shift > 180 $cm^{-1}$ and the filter cut-off at around 100 $cm^{-1}$, the following analysis will be limited to the region 100–400 $cm^{-1}$. Moreover, one should note that the observed intensity values at Raman shift < 150 $cm^{-1}$ are not completely reliable due to the closeness to the filter cut-off, thus preventing us from performing quantitative relative analyses. When the 24 h-loaded systems are analysed (Fig. 3c), solid/molecular iodine is clearly detectable at 180 $cm^{-1}$, as well as a band at 163 $cm^{-1}$ presumably related to the interaction of iodine with the oxygen donors of glycerol (whose overtone is located at 325 $cm^{-1}$)[27]. Essentially, this implies that two uptake mechanisms occur: on the one hand, $I_2$ is physically stored in its molecular form within the DESs; on the other hand, $I_2$ chemically interacts with glycerol molecules ($-O\text{---}I_2$). This interaction occurs between Iodine and an oxygen atom bearing a partially negative charge (*i.e.* $-O^{\delta-}$); as highlighted by some of us in a recent paper[28] dealing with glycerol/halogen salts-based DESs, the establishment of a highly directional $X^-$-HOGly hydrogen bond moved the hydrogen quite close to the halogen atom, thus leading to the formation of a glycerolate-like species. As expected, the polarisation of the hydrogen bond is directly related to the electronegativity of the halogen, with the $Cl^-$ being more electron-attractive than the $I^-$. In other words, the formation of the $Cl^-$-HOGly hydrogen bond leads to the formation of a glycerolate-like species bearing a more negative partial charge, giving rise to the higher effectiveness in iodine uptake ensured by ChCl/Gly 1:2 DES (see Fig. 1b).

For being proposed as iodine sponges, besides the retention of radioactive iodine, these systems must be able to release it under a certain stimulation. For this purpose, the three samples were heated at 100 °C for 30 min, aiming to unload them. After this treatment, the mixtures were investigated by means of Raman spectroscopy. As a result, DES1, DES2, and DES3 lost 76%, 72% and 78% of the iodine previously captured and leading to a residual Iodine load as high as $0.95 \, g \, g^{-1}$, $0.79 \, g \, g^{-1}$ and $0.61 \, g \, g^{-1}$ respectively, in reasonable agreement with TGA data (Fig. 2). The consistent iodine unloading directly reflects on I-speciation within the DES: as shown in Fig. 3d, the Raman spectra do not present any peak ascribable to physically absorbed $I_2$ nor to the $-O\text{---}I_2$ interaction (as well as its overtone). On the other hand, three bands ($\approx$168, 143, and 109 $cm^{-1}$), tentatively ascribable to $I_5^-$ (Table 1), could be highlighted in all the samples. The latter is likely formed after heating, probably due to the release and then the rearrangement of residue iodine species under the thermal process. One should note that the formation of $I_5^-$ in ChCl-based DESs must imply a halogen exchange and/or the evaporation of Cl-containing species. More detailed analyses are currently undergoing to disclose this assignment, as the formation of more exotic species (*e.g.* $-O\text{---}I_{2n}\text{---}O$), which do not directly involve a chemical interaction with $X^-$, could not be excluded. The occurrence of this type of interactions is further suggested by the analyses of the Raman spectrum of glycerol after 24 h of iodine uptake (8 ms%), which reveals bands associated with various polyiodide species (see Supplementary Fig. 3). Notably, the iodine speciation in pure glycerol closely resembles that observed in DES mixtures following thermal stress, displaying three bands attributable to $I_5^-$. Furthermore, upon heating, a rearrangement of iodide species is occurring, this being indicated by a partial iodine release ($\approx$5 ms%) and the corresponding increase in the relative contribution of $I_3^-$. Focusing on the

practical application, it should be pointed out that all DESs are able to store a significant amount of molecular iodine and then release almost 80% of it under relatively mild conditions (*i.e.* 100 °C for 30 min).

In summary, the results discussed throughout the present communication pinpoint the applicability of Deep Eutectic Solvents as cost-effective and environmentally friendly mixtures to uptake significant amounts (up to $4 \, g \, g^{-1}$) of molecular iodine, which could be then almost completely reversibly released under mild conditions. These findings lay the foundation for further engineering of these mixture to fully disclose their potential. Moreover, the exploitation of Raman spectroscopy enables the determination of the I-containing species giving valuable insights into the possible uptake mechanisms and paving the way for further engineering of glycerol-based molecular solvents. Additional studies are actually undergoing to assess the potential for recovering the portion of iodine still trapped within the DES, as well as to explore the feasibility of exploiting these systems directly as I-based electrolytes in energy storage and harvesting applications.

## Methods

### Materials

Choline chloride (Sigma-Aldrich, ≥98%, CAS No: 67-48-1), choline iodide (TCI, >98%, CAS No: 17773-10-3), glycerol (Sigma-Aldrich, ≥99.5%, CAS No: 56-81-5), molecular iodine (Sigma-Aldrich, ≥99.8%, CAS No: 7553-56-2). Before use, glycerol and choline salts were vacuum-dried to remove spurious water until a constant weight was obtained.

### DESs formulation

Three glycerol-based DESs were prepared, namely choline chloride and choline iodide in combination with glycerol in molar ratios 1:2 and 1:3. The mixtures were formulated under stirring at 40 °C until a homogenous liquid appeared. In the whole document, DES1, DES2, and DES3 refer to deep eutectic solvents made of choline chloride/glycerol 1:2, choline chloride/glycerol 1:3, and choline iodide/glycerol 1:3, respectively.

### Iodine uptake experiment

Four open small glass vials containing 10 mg of DES1, DES2, DES3, and glycerol were inserted in a sealed wide-mouth glass jar filled with an excess amount of molecular iodine to ensure the saturation of the environment. The whole was then kept at 80 °C in an oven. At different contact times, the wide-mouth jar was removed from the oven and cooled down to room temperature. Each time, the small glass vial containing the absorbents was weighed and then placed back into the wide-mouth jar to continue the $I_2$ capturing, until its weight reached a steady value. Supplementary Fig. 4 are images of the experimental setup used for the $I_2$ uptake experiment.

Iodine uptake capacity was calculated using the following equation:

$$U = \frac{m_2 - m_1}{m_1}$$

where $U$ represents the iodine uptake, $m_1$, and $m_2$ are the mass of absorbents before and after being exposed to iodine vapour. Each value obtained from the uptake was normalised by the initial mass of the used absorbent. Experiments were performed three times with consistent results, and the arithmetic mean of $U$ is reported for each mixture.

### Thermal analysis

TGA analyses were recorded with a TA instruments Q600 thermo-balance under 100 mL $min^{-1}$ air flow with temperature ramps of 5 °C $min^{-1}$. For determining the thermal stability of pure glycerol and DESs the temperature was sweep from 30 °C to 325 °C, while to investigate weight losses of iodine-loaded DESs following thermal stress the temperature was sweep from 30 °C to 180 °C.

 

## Raman investigation

Raman spectra were acquired using a laser with $\lambda = 785$ nm, associated with the Renishaw Raman Microscope 1000 instrument, after calibrating it with monocrystalline silicon, by applying 0.5% laser power with a 5x NIR objective lens for 10 acquisitions, each lasting 20 s. For all the mixtures, the region $100\text{--}1800\ cm^{-1}$ was examined, with a resolution of $2\ cm^{-1}$.

## Data availability

All data that support the findings of this study are included in the manuscript and/or Supplementary Information files. Those data are also available from the corresponding authors upon reasonable request.

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

## Acknowledgements

This study has received support from GENESIS (Green Electrolyte and Biomass-derived Electrodes for Sustainable Electrochemical Storage Devices - 2022ZZHS7L) project funded by the Ministero dell'Università e della Ricerca within the PRIN 2022 program (CUP: D53D23009250001), from the research project "nuovi Concetti, mAteriali e tecnologie per l'iNtegrazione del fotoVoltAico negli edifici in uno scenario di generazione diffuSa" (CANVAS) funded by the Italian Ministry of the Environment and the Energy Security through the Research Fund for the Italian Electrical System (type-A call, published on G.U.R.I. n. 192 on 18-08-2022). The authors acknowledge support from Project CH4.0 under the MUR program "Dipartimento di Eccellenza 2023–2027" (CUP D13C22003520001). S.M. acknowledges Italian Ministry of Foreign Affairs and International Cooperation (MAECI) for funding the "Cameroon scholarship 2023".

## Author contributions

D.M. performed the data analyses, collected the data, edited the figures, and contribute to writing the first draft of the manuscript. S.M. collected the data and contributed to writing the first draft of the manuscript. S.N. and C.P. validated the findings and revised the manuscript. C.B. supervised the project, validated the findings, and revised the manuscript. A.D. and M.B. contributed to data collection, data analysis, methodology development, manuscript revision, and project administration.

## Competing interests

The authors declare no competing interests.
