## [Transparent Peer Review file · Communications Chemistry]

Glycerol-based deep eutectic solvents for efficient and reversible iodine uptake from vapour phase

Corresponding Author: Professor Matteo Bonomo

Version 0:

Reviewer comments:

Reviewer #1

(Remarks to the Author)

Bonomo and co-workers describe the development of cholinium halide/glycerol deep eutectic solvents (DES) as effective I₂ sorbents, being the first time that they were directly used for I₂ gas-phase capture. The captured I₂ can be partially released under relatively mild conditions, showcasing the DES's potential for semi-reversible I₂ storage. The Raman spectroscopy study offers detail on the species involved in the capture process, which serves as a basis for further investigations. In addition, the obtained results show reproducibility and consistency when analyzing data from different techniques. Considering the low cost and broad availability of the DES components, these systems could be of interest not only to the academic community but also for their potential application in the field.

Although the experimental results thoroughly support the specific statements presented, we recommend a few additional straightforward experiments to bolster the data.

The stability of the system once the I₂ is captured in the DES is a relevant parameter that has been measured by others in related studies (Green Chem., 2016,18, 2522-2527 and ACS Appl. Mater. Interfaces 2025, 17, 8382–8393). It would be interesting to analyze the behavior of these mixtures regarding their stability over time.

Considering the gravimetric methodology and the inherent hydrophilicity of the DESs, it is important to address potential DES hydration in the I₂ uptake experiment. Please specify in the methodology how hydration was prevented.

TGA analysis of pure DES is recommended to demonstrate the DES's thermal stability and rule out decomposition during I₂ release. In the case of glycerol, the authors claim (lines 139-140) that the sudden drop in mass around 140 degrees could be ascribed to the degradation of the glycerol. The authors should provide a supporting reference.

Although iodine capture and TGA studies have been conducted on both the DES and glycerol, Raman spectroscopy has only been performed on the DES. Including Raman results for glycerol would enhance the study; if these were omitted, a justification for their absence should be provided.

Minor comments:

There is related literature about DES for iodine capture that is missing (Journal of Molecular Liquids 412 (2024) 125883). Please add this reference.

Figure 1-b is difficult to analyze at first glance due to the time unit used. Consistent with Figure 1c, hours should be used. Similarly, the plateau data in line 108 should be expressed in hours.

A 'B' example is not depicted in Figure 1c. To maintain consistency, the figure caption should be corrected, and the letter labels in the figure should be revised.

As the experiments were done in triplicate, the standard deviation should be shown for a better comparison of the uptake results.

Reviewer #2

(Remarks to the Author)

I co-reviewed this manuscript with one of the reviewers who provided the listed reports. This is part of the Communications Chemistry initiative to facilitate training in peer review and to provide appropriate recognition for Early Career Researchers who co-review manuscripts.

Reviewer #3

(Remarks to the Author)

The efficient capture and recovery of radioactive iodine (I₂) is critical for nuclear safety and environmental protection. Glycerol (Gly)-based deep eutectic solvents (DESs) have recently emerged as promising and novel solvents for iodine uptake from the vapor phase. The iodine uptake performance varies depending on the choice of cholinium salt (Ch⁺) as the

hydrogen bond acceptor (either iodide or chloride) and its relative ratio with glycerol (1:2 or 1:3). Among these, the ChCl:Gly (1:2) system exhibits the best performance, reaching up to $4 \text{ g}\cdot\text{g}^{-1}$ after 24 hours. Furthermore, the captured iodine can be effectively released (approximately 80%), demonstrating the potential of DESs as iodine sponges. Raman spectroscopy has been employed to confirm the speciation of iodine-based species within the DES, which opens up avenues for further engineering of these systems. These results highlight Gly-based DESs as a sustainable, low-cost solvent for efficient and semi-reversible iodine capture.

Before publication, the following points should be addressed:

1 The first work to employ ionic liquids for iodine capture is found in *Physical Chemistry Chemical Physics*, 16 (11), 5071-5075, and should be cited.

2 In Figures 1b and 1c, time is expressed in minutes and hours separately. For consistency, it would be preferable to use hours for both.

3 Is it possible to quantify iodine species based on the area of these species in Figure 3, or through other methods?

4 The iodine capture rate appears to depend on the viscosity of the DESs. Please provide the viscosity data for the DESs and discuss this point. A relevant reference for this discussion is *Physical Chemistry Chemical Physics*, 24 (42), 26029-26036.

Version 1:

Reviewer comments:

Reviewer #1

(Remarks to the Author)

I consider that this new version of the manuscript contains substantial improvements compared to the initial version, and that the authors have satisfactorily addressed each of the points raised in the previous review round. Therefore, I am satisfied with this revised version.

Reviewer #2

(Remarks to the Author)

I co-reviewed this manuscript with one of the reviewers who provided the listed reports. This is part of the Communications Chemistry initiative to facilitate training in peer review and to provide appropriate recognition for Early Career Researchers who co-review manuscripts.

Reviewer #3

(Remarks to the Author)

I suggest to accept the revised manuscript for publication.

REVIEWER REPORT(S):

Referee #1 and #2

Bonomo and co-workers describe the development of cholinium halide/glycerol deep eutectic solvents (DES) as effective I₂ sorbents, being the first time that they were directly used for I₂ gas-phase capture. The captured I₂ can be partially released under relatively mild conditions, showcasing the DES's potential for semi-reversible I₂ storage. The Raman spectroscopy study offers detail on the species involved in the capture process, which serves as a basis for further investigations. In addition, the obtained results show reproducibility and consistency when analyzing data from different techniques. Considering the low cost and broad availability of the DES components, these systems could be of interest not only to the academic community but also for their potential application in the field.

Although the experimental results thoroughly support the specific statements presented, we recommend a few additional straightforward experiments to bolster the data.

We are thankful to the referee #1 and #2 for the general positive evaluation of our manuscript and for the very meaningful suggestions. We specifically commented on all the issues raised and we modified the manuscript accordingly. After the modifications, we are fairly confident that the manuscript could be accepted for publication in Communications Chemistry.

Major	
The stability of the system once the I ₂ is captured in the DES is a relevant parameter that has been measured by others in related studies (Green Chem., 2016,18, 2522-2527 and ACS Appl. Mater. Interfaces 2025, 17, 8382–8393). It would be interesting to analyze the behavior of these mixtures regarding their stability over time.	We agree on this point. Indeed, the stability of the I₂-loaded DES is quite important, especially envisaging an industrial application. We are currently working on this, also using Raman to determine the speciation of I-containing species. In the submitted version of the manuscript, we did not explicitly discuss this point because the main focus was the presentation of DES as effective iodine sponges. However, in the meantime, we have continuously monitored the I₂-loaded DESs, and they are pretty stable over time under ambient conditions (i.e. an open vial at RT and atmospheric pressure). As a matter of fact, after an initial slight decrease in weight (i.e. 3-6%, likely due to the evaporation of the physically sorbed iodine at the DES surface), it maintains a relevant constancy over 42 days of exposure. The following short paragraph has been added in the main text to better discuss this point: “Once reached the plateau of the I₂ uptake, all the samples showed a quasi-solid gel-like behaviour, which is even more evident when the samples are

	cooled down to RT. This is fairly ascribable to the establishment of a new iodine-based interaction between DES moieties (see Raman section). The high-energy nature of the latter leads to extremely stable I₂-loaded sponges which lose only an amount between 3% and 6% of mass after more than one month (i.e. 42 days) of storage under ambient conditions. As shown in Supplementary Figure 1, almost all the weight is lost within the first 4 hours of storage, likely due to the evaporation of the loosely interacting I₂ at the DES exposed surface. If on the one hand, the establishment of highly energetic interactions is surely an added value in terms of long-term stability toward safe and reliable industrial application, it would also require some energy to allow an effective de-loading of the iodine (vide infra).”
Considering the gravimetric methodology and the inherent hydrophilicity of the DESs, it is important to address potential DES hydration in the I₂ uptake experiment. Please specify in the methodology how hydration was prevented.	We acknowledge that water absorption may influence gravimetric measurements. However, as shown in a previous study (Phys. Chem. Chem. Phys., 2019, 21, 2601), water uptake can theoretically reach a maximum of 10% by mass, which in our case would account for only 2–3% of the total uptake, an amount that can be considered negligible (as it would be within the experimental error). Additionally, given that the sample bottles were saturated with vapours of iodine and based on the observed kinetic behaviour, the likelihood of significant water absorption is greatly reduced. Finally, the absence of any detectable water-related Raman features further confirms its minimal contribution to the measurements. The following sentence has been added to the revised version of the text to clarify this point: “Although water absorption may influence gravimetric measurements, its impact is considered negligible in the present case. Based on previous findings [21], the theoretical maximum water absorption capacity for these kinds of systems is approximately 10% by mass, which would represent only 2–3% of the total uptake observed (i.e. a value within the experimental uncertainty). Furthermore, the saturation of sample containers with iodine vapours and the kinetic behaviour observed suggest limited room for significant water absorption.”

TGA analysis of pure DES is recommended to demonstrate the DES's thermal stability and rule out decomposition during I₂ release. In the case of glycerol, the authors claim (lines 139-140) that the sudden drop in mass around 140 degrees could be ascribed to the degradation of the glycerol. The authors should provide a supporting reference.	We acknowledge the reviewer for the very valuable comment. Following on from the suggestion we added the thermal stability of the pure mixtures in the Supporting Information (Supporting Figure 1). To clarify the point, the sentence has been rephrased as follows: “The sudden drop at T > 140 °C is ascribable to the degradation of the glycerol, which occurs between 200 °C and 213 °C, both in its pure form and within the mixtures (Supplementary Figure 2).”
Although iodine capture and TGA studies have been conducted on both the DES and glycerol, Raman spectroscopy has only been performed on the DES. Including Raman results for glycerol would enhance the study; if these were omitted, a justification for their absence should be provided.	We thank the reviewers for this comment, and we apologize for not having included a discussion about the Raman spectra of pure glycerol, already reported in the supporting information. A short paragraph concerning it has been added in the main text. “The occurrence of this type of interactions is further suggested by the analyses of the Raman spectrum of glycerol after 24 hours of iodine uptake (8 ms%), which reveals bands associated with various polyiodide species (see Supplementary Figure 3). Notably, the iodine speciation in pure glycerol closely resembles that observed in DES mixtures following thermal stress, displaying three bands attributable to I₅⁻. Furthermore, upon heating, a rearrangement of iodide species is occurring, this being indicated by a partial iodine release (≈5 ms%) and the corresponding increase in the relative contribution of I₃⁻.”
Minor	
There is related literature about DES for iodine capture that is missing (Journal of Molecular Liquids 412 (2024) 125883). Please add this reference.	We thank the reviewers for the valuable suggestion. For completeness we have also integrated two other works in addition to the one suggested.
Figure 1-b is difficult to analyze at first glance due to the time unit used. Consistent with Figure 1c, hours should be used. Similarly, the plateau data in line 108 should be expressed in hours.	We thank the reviewers for the valuable suggestion. All data has been standardized into hours.
A 'B' example is not depicted in Figure 1c. To maintain consistency, the figure caption should be	We thank the reviewers for this comment, and we apologize for the extra label. The mistake has been fixed.

corrected, and the letter labels in the figure should be revised.	
As the experiments were done in triplicate, the standard deviation should be shown for a better comparison of the uptake results.	We acknowledge the reviewer for the very valuable comment. Following on from the suggestion, in Figure 1b, we added the statistical variation for each experimental point (indeed, they were omitted in the original version being lower than 5% on average). Moreover, two tables summarizing all the experimental data have been added in the supporting information file (see Supplementary Tables 1 and 2).

Referee #3

The efficient capture and recovery of radioactive iodine (I_2) is critical for nuclear safety and environmental protection. Glycerol (Gly)-based deep eutectic solvents (DESs) have recently emerged as promising and novel solvents for iodine uptake from the vapor phase. The iodine uptake performance varies depending on the choice of cholinium salt (Ch^+) as the hydrogen bond acceptor (either iodide or chloride) and its relative ratio with glycerol (1:2 or 1:3). Among these, the $ChCl:Gly$ (1:2) system exhibits the best performance, reaching up to $4 \text{ g}\cdot\text{g}^{-1}$ after 24 hours. Furthermore, the captured iodine can be effectively released (approximately 80%), demonstrating the potential of DESs as iodine sponges. Raman spectroscopy has been employed to confirm the speciation of iodine-based species within the DES, which opens up avenues for further engineering of these systems. These results highlight Gly-based DESs as a sustainable, low-cost solvent for efficient and semi-reversible iodine capture.

Before publication, the following points should be addressed:

We thank the referee #3 for the very positive feedback on our manuscript. We carefully considered all the comments raised and modified the manuscript accordingly. We are fairly convinced that the revised version is suitable for publication in Communications Chemistry.

The first work to employ ionic liquids for iodine capture is found in Physical Chemistry Chemical Physics, 16 (11), 5071-5075, and should be cited.	We thank the reviewer for the valuable suggestion, and the reference has now been incorporated into the main text.
In Figures 1b and 1c, time is expressed in minutes and hours separately. For consistency, it would be	We thank the reviewer for the valuable suggestion. All data has been standardized into hours.

preferable to use hours for both.	
Is it possible to quantify iodine species based on the area of these species in Figure 3, or through other methods?	Yes, it could be possible; however, as noted in the main text, “the observed intensity values at Raman shifts below 150 cm⁻¹ are not entirely reliable due to their proximity to the filter cut-off,” which prevents us from carrying out accurate quantitative analyses in this region. Furthermore, a reliable quantitative Raman analysis would require knowledge of the Raman cross-sections for each species (same for absorption coefficients in UV-Vis spectroscopy), which falls beyond the scope of this communication. This aspect will certainly be addressed in future and more focused studies, particularly once we will gain access to the low frequency region of the Raman spectrum.
The iodine capture rate appears to depend on the viscosity of the DESs. Please provide the viscosity data for the DESs and discuss this point. A relevant reference for this discussion is Physical Chemistry Chemical Physics, 24 (42), 26029-26036.	We kindly disagree with the reviewer on this point. Although a dependence of I₂ uptake on viscosity could not be excluded a-priori, it seems far to be straightforward. Indeed, based on our previous publication (see Bonomo, M. et al. Journal of Molecular Liquids, 2020, 319, 114292), pure glycerol and the 1:3 ChCl:Gly mixture showed the highest and the lowest viscosity, respectively, with the 1:2 ChCl:Gly mixture showing intermediate values (closer to 1:3 ChCl:Gly, indeed). However, this trend seems not to be replicated in the Iodine uptake values, with 1:2 ChCl:Gly showing the highest amount. As such, even if it could be speculated that the much higher viscosity of pure glycerol has a negative effect in I₂ diffusion (and uptake) throughout the solvent, we are strongly convinced (supported by Raman spectroscopy data) that the main role in enhancing iodine uptake is placed by the formation of partially negatively charged oxygen moieties following on from the addition of ChCl and the establishment of DES(-like) HBA/HBD interactions.